# Hypothyroidism After Hemithyroidectomy: A Retrospective Analysis of Temporal Trends and Key Risk Factors

**DOI:** 10.3390/jcm14030919

**Published:** 2025-01-30

**Authors:** Alaa Safia, Rabie Shehadeh, Adi Sharabi-Nov, Yaniv Avraham, Ohad Ronen, Shlomo Merchavy

**Affiliations:** 1Department of Otolaryngology, Head and Neck Surgery, Ziv Medical Center, Safed 1311001, Israel; shlomo.m@ziv.health.gov.il; 2The Azieli Faculty of Medicine, Bar Ilan University, Ramat Gan 5290002, Israel; rshehadeh1@gmail.com (R.S.); adi_nov@hotmail.com (A.S.-N.); yaniv@orhaganuz.com (Y.A.); ohadr@gmc.gov.il (O.R.); 3Department of Otolaryngology, Head and Neck Surgery, Galilee Medical Center, Nahariya 2210001, Israel

**Keywords:** hemithyroidectomy, hypothyroidism, incidence rate, determinants

## Abstract

**Background/Objectives**: Post-hemithyroidectomy hypothyroidism is a recognized complication, though its incidence and risk factors remain variably reported. This study aimed to determine the incidence of hypothyroidism post-hemithyroidectomy, identify associated risk factors, and assess temporal changes in thyroid hormone levels. **Methods**: This retrospective analysis examined the records of 192 euthyroid patients who underwent hemithyroidectomy between January 2019 and May 2023. Thyroid function was assessed preoperatively and at 1, 6, and 12 months postoperatively. Kaplan–Meier survival analysis and Cox proportional hazards regression were used to evaluate the time to hypothyroidism onset and identify significant predictors. **Results**: At 1 month postoperatively, a higher incidence was observed in elderly patients (>65 years; 31.11%) and those aged 56–65 years (29.54%), compared to younger age groups (7.69% in patients aged 18–35 years). The incidence decreased over time, with 14.06% and 10.94% of patients being hypothyroid at 6 and 12 months, respectively. Severe obesity was associated with the highest hypothyroidism rates across all follow-up periods, particularly at 1 month (50.00%). Elevated BMI was also associated with increased risk, particularly in overweight patients (HR = 2.368, 95% CI 1.016–5.523). Patients undergoing left-sided hemithyroidectomy had a higher incidence of hypothyroidism at 12 months compared to right-sided hemithyroidectomy patients (15.63% vs. 6.25%, *p* = 0.037). Cox regression analysis confirmed diabetes and BMI (overweight) as significant predictors of hypothyroidism. **Conclusions**: Hypothyroidism is a common complication following hemithyroidectomy, particularly in elderly and overweight patients. The significant early onset of thyroid dysfunction underscores the need for routine postoperative monitoring, especially within the first year.

## 1. Introduction

Hypothyroidism is a well-documented, but often underappreciated, complication following hemithyroidectomy, the surgical removal of one thyroid lobe and the isthmus [1,2]. While the procedure is generally regarded as less likely to result in thyroid dysfunction compared to total thyroidectomy, emerging evidence suggests that a substantial proportion of patients develop hypothyroidism after hemithyroidectomy, even in the absence of pre-existing thyroid disease [1,3,4,5]. The postoperative decline in thyroid function can vary, ranging from transient, subclinical hypothyroidism, to overt, long-term thyroid insufficiency requiring hormone replacement therapy [3].

The development of hypothyroidism following hemithyroidectomy has important clinical implications. Even mild thyroid dysfunction can have a profound impact on quality of life, leading to symptoms such as fatigue, weight gain, and cognitive impairment [6,7]. Moreover, subclinical hypothyroidism has been associated with increased cardiovascular risk, particularly in older adults and individuals with metabolic syndrome [8,9]. Identifying patients at risk of postoperative hypothyroidism and implementing appropriate monitoring and management strategies are therefore critical for optimizing outcomes in this patient population.

Previous studies have reported widely varying rates of hypothyroidism following hemithyroidectomy, with incidences ranging from 15% to 30%, depending on patient demographics, preoperative thyroid function, and surgical technique [1,2,3,4,5]. While several risk factors have been proposed, including age, body mass index (BMI), and the extent of thyroid resection [1,3,4,10], a clear consensus has not yet been established. The role of surgical laterality—whether the right or left lobe is removed—also remains a subject of debate, with some studies suggesting a potential influence of anatomical variations on postoperative thyroid function [11].

Despite the clinical importance of hypothyroidism following hemithyroidectomy, current guidelines offer limited recommendations regarding postoperative follow-up and the timing of thyroid function testing [12]. Given the potential for both early and late onset of hypothyroidism, understanding the temporal patterns of thyroid dysfunction and identifying key predictors of hypothyroidism are essential for guiding postoperative care.

In this retrospective cohort study, we sought to determine the incidence and timing of hypothyroidism following hemithyroidectomy in a well-defined population of euthyroid patients. We further aimed to identify patient-specific risk factors, including age, BMI, and surgical laterality, that may predict the development of postoperative hypothyroidism. By providing a comprehensive analysis of the temporal changes in thyroid function and the factors influencing hypothyroidism risk, our study aims to fill critical gaps in the existing literature and inform future guidelines for the management of patients undergoing hemithyroidectomy.

## 2. Materials and Methods

### 2.1. Study Design and Population

This retrospective observational study was conducted to evaluate the incidence and risk factors of hypothyroidism following hemithyroidectomy in euthyroid patients. We analyzed medical records from January 2019 to May 2023, drawing from two major tertiary care centers—Ziv Medical Center and Western Galilee Medical Center. Only patients with normal preoperative thyroid function were included, while those with a history of thyroid dysfunction or previous thyroid surgery were excluded, to allow for a precise assessment of predictors of postoperative hypothyroidism. This study was conducted in accordance with ethical guidelines and received approval from the Institutional Review Board (0035-21-ZIV), ensuring adherence to the principles outlined in the Helsinki Declaration. Stringent measures were taken to ensure methodological rigor and consistency.

### 2.2. Eligibility Criteria

The study cohort comprised all adult patients (age ≥ 18 years) who underwent hemithyroidectomy between January 2019 and May 2023. Eligible patients met the following criteria:Normal thyroid function before surgery, confirmed by biochemical assays;No prior thyroid surgery;Hemithyroidectomy and isthmectomy performed, with preservation of the contralateral lobe.

Patients with preoperative hypo- or hyperthyroidism, or those who underwent total or subtotal thyroidectomy, were excluded. The primary outcome was the development of hypothyroidism, with secondary outcomes assessing the requirement for thyroid hormone supplementation. Data on patient demographics, clinical characteristics, and surgical details were also collected.

### 2.3. Measurements, Definitions, and Follow-Up

Thyroid function was assessed at multiple time points, including preoperatively and at follow-up visits 1 month, 6 months, and 12 months postoperatively. Subclinical hypothyroidism was defined as a serum thyroid-stimulating hormone (TSH) level of 4.20–10 µIU/mL with normal free T4 levels in asymptomatic patients, as per the American Thyroid Association/American Association of Clinical Endocrinologists (ATA/AACE) 2012 guidelines [13]. Overt hypothyroidism was defined by elevated TSH levels (>10 µIU/mL) combined with a low free T4 level. Both conditions were considered as “hypothyroidism” for this study. Thyroid hormone replacement therapy was initiated in patients with TSH >10 µIU/mL or those with a low free T4 level, particularly if accompanied by clinical symptoms such as fatigue, weight gain, or depression. These thresholds were based on clinical guidelines and institutional protocols. The time to hypothyroidism onset was defined as the interval between the date of surgery and the first documented biochemical evidence of hypothyroidism. To ensure consistency and reduce bias, two independent members of the research team reviewed patient symptoms and laboratory data. Symptoms were identified through structured screening questions during clinical visits and supplemented by patient or caregiver reports.

Serum TSH and thyroxine (T4) levels were measured preoperatively and at 1, 6, and 12 months postoperatively. Thyroid function tests were performed using electrochemiluminescence immunoassay (ECLIA) technology on the Cobas e immunoassay analyzer (Roche Diagnostics GmbH). The assays used included the Roche kits Ref. 06437281 for T4 and Ref. 08429324 for TSH, with normal reference ranges corresponding to the 2.5th and 97.5th percentiles. The onset of hypothyroidism was defined as the time from surgery to the first documented abnormal TSH level, measured in months.

### 2.4. Statistical Analysis

All statistical analyses were performed using Stata [version 18, StataCorp LLC, USA]. Continuous variables were expressed as means with standard deviations (SDs), while categorical variables were presented as counts and percentages. For baseline comparisons, continuous variables were compared using independent samples *t*-tests or analysis of variance (ANOVA) for normally distributed data, while categorical variables were compared using the chi-square test or Fisher’s exact test, where appropriate.

Changes in thyroid function over time (T3, T4, and TSH levels) were analyzed using repeated measures analysis of variance (ANOVA). Post hoc pairwise comparisons between preoperative results and results at each follow-up time point (1 month, 6 months, and 12 months) were conducted using Bonferroni-adjusted *t*-tests. The mean changes in T3, T4, and TSH levels over time were reported with 95% confidence intervals (CIs), and statistical significance was set at a two-sided *p*-value of less than 0.05.

The incidence of hypothyroidism was assessed at 1, 6, and 12 months postoperatively. Comparisons of hypothyroidism rates across patient subgroups (age, gender, BMI, and surgical laterality) were performed using chi-square or Fisher’s exact tests, as appropriate. Kaplan–Meier survival curves were generated to assess the time to development of hypothyroidism, stratified by baseline characteristics, with log-rank tests used to compare survival curves between subgroups.

To identify factors associated with the development of hypothyroidism, both unadjusted and multivariable Cox proportional hazards regression models were used. Variables included in the multivariable model were selected based on clinical relevance and statistical significance in univariate analysis (*p* < 0.10). Hazard ratios (HRs) with 95% confidence intervals were reported, and statistical significance was set at *p* < 0.05. The proportional hazards assumption was tested using Schoenfeld residuals. All analyses were two-sided, and *p*-values <0.05 were considered statistically significant, unless otherwise specified.

## 3. Results

### 3.1. Baseline Clinicodemographic Data

The study cohort included a total of 192 patients who underwent hemithyroidectomy (Table 1). The mean age was 53.15 years (SD, 14.33), with 26 patients (13.54%) classified as young adults (18–35 years), 77 (40.10%) as middle-aged (36–55 years), 44 (22.92%) as older adults (56–65 years), and 45 (23.44%) as elderly (>65 years). The majority of patients were female (154, 80.21%), with only 38 (19.79%) males. The mean body mass index (BMI) was 28.32 (SD, 5.96), with 49 patients (29.52%) classified as having normal weight, 59 (35.54%) as overweight, 50 (30.12%) as obese, and 8 (4.82%) as severely obese.

Regarding family history of thyroid cancer, only seven patients (3.66%) had a positive family history. Hemithyroidectomy was equally distributed between the right (96 patients, 50%) and left (96 patients, 50%) sides. The primary indications for surgery were goiter (132 patients, 68.75%), papillary thyroid carcinoma (PTC) (29 patients, 15.10%), and follicular adenoma (26 patients, 13.54%).

### 3.2. Changes in Thyroid Parameters over Time

Preoperatively, the mean triiodothyronine (T3), thyroxine (T4), and thyroid-stimulating hormone (TSH) levels were 4.71 ng/mL (SD, 0.64), 15.17 µg/dL (SD, 1.94), and 2.00 mIU/L (SD, 1.26), respectively. There was a significant reduction in T4 levels at 1 month compared to preoperative values (−0.48 µg/dL, 95% CI [−0.89, −0.07]; *p* = 0.023). By 6 and 12 months, the changes in T4 levels were not statistically significant (*p* = 0.064 and *p* = 0.436, respectively). T3 levels did not show significant changes across the follow-up periods. For TSH, a statistically significant increase was observed at 1 month postoperatively (mean increase of 0.90 mIU/L, 95% CI [0.54, 1.25]; *p* < 0.001). By 6 months, TSH levels decreased slightly, but remained elevated compared to baseline (*p* = 0.036). By 12 months, there was no significant change in TSH levels (*p* = 0.341). The overall changes in T3, T4, and TSH are summarized in Figure 1 and Table 2.

### 3.3. Temporal Trends of Post-Hemithyroidectomy Hypothyroidism

At 1 month postoperatively, 41 patients (21.35%) developed hypothyroidism, while 151 patients (78.65%) remained euthyroid. Among those who developed hypothyroidism, the incidence was highest in patients > 65 years (31.11%), followed by those aged 56–65 years (29.54%). In contrast, younger patients aged 18–35 years had the lowest incidence (7.69%). At 6 months, 27 patients (14.06%) were hypothyroid, with a continued higher incidence in older age groups. By 12 months, 21 patients (10.94%) remained hypothyroid, with the elderly (>65 years) showing a rate of 17.78%. This temporal trend highlights that older age groups consistently exhibit higher hypothyroidism rates compared to younger cohorts.

Regarding BMI, patients with severe obesity demonstrated the highest incidence of hypothyroidism across all follow-up periods, with rates of 50.00% at 1 month, 25.00% at 6 months, and 12.50% at 12 months. Patients with normal weight exhibited a lower incidence (14.28% at 1 month). However, the association between BMI and hypothyroidism was not statistically significant (*p* > 0.05 at all time points).

No significant differences were observed in hypothyroidism incidence based on gender or family history of thyroid cancer. Surgical laterality, however, showed a significant difference at 12 months, with patients who had undergone left-sided hemithyroidectomy having a higher incidence of hypothyroidism (15.63%) compared to those who had undergone right-sided hemithyroidectomy (6.25%; *p* = 0.037). The details are provided in Table 3.

### 3.4. Predictors of Time to Hypothyroidism Post-Hemithyroidectomy

The Kaplan–Meier curves stratified by age, gender, BMI, and surgical laterality are displayed in Figure 2. The log-rank test demonstrated a statistically significant difference in hypothyroidism-free survival between age groups (*p* = 0.013) and surgical laterality (*p* = 0.037). Cox proportional hazards analysis revealed that age and BMI were significant predictors of hypothyroidism (Table 4).

In the adjusted model, being overweight (HR 2.368, 95% CI [1.016, 5.523], *p* = 0.046) significantly increased the risk of hypothyroidism, while severe obesity was not a significant predictor (HR 3.185, 95% CI [0.906, 11.201], *p* = 0.071). Age >65 years was also associated with an increased risk, although this was not statistically significant in the adjusted model (HR 1.500, 95% CI [0.490, 4.594], *p* = 0.478). Diabetes mellitus was identified as a protective factor (adjusted HR 0.379, 95% CI [0.147, 0.975], *p* = 0.044). Laterality showed a trend towards higher risk for hypothyroidism with left-sided surgery (adjusted HR 1.682, 95% CI [0.903, 3.134], *p* = 0.102), but this did not reach statistical significance in the multivariable model. The full Cox regression results are presented in Table 4.

## 4. Discussion

In this retrospective cohort study of euthyroid patients undergoing hemithyroidectomy, we identified several key findings regarding the incidence, timing, and risk factors associated with the development of hypothyroidism postoperatively. Our results not only confirm previous observations regarding the frequency of hypothyroidism after hemithyroidectomy, but also provide novel insights into the temporal changes in thyroid function and the impact of patient-specific factors on hypothyroidism risk.

The incidence of hypothyroidism in our cohort was substantial, with 21.35% of patients developing hypothyroidism at 1 month postoperatively. While the majority of cases were transient, the persistence of hypothyroidism in 10.94% of patients at 12 months indicates that long-term thyroid dysfunction remains a significant concern following hemithyroidectomy. These findings align with previous studies that have reported rates of postoperative hypothyroidism ranging from 15% to 30%, underscoring the importance of ongoing surveillance in this patient population [1,3,4]. The early onset of hypothyroidism observed at 1 month in a substantial proportion of patients may reflect the initial stress and adaptive response of the remaining thyroid lobe. Although a significant proportion of patients recovered thyroid function by 6 and 12 months, the persistence of hypothyroidism in a notable subset suggests that some patients may experience permanent thyroid insufficiency [14]. This highlights the need for individualized follow-up strategies and timely intervention with thyroid hormone replacement where appropriate.

Our study identified several important predictors of postoperative hypothyroidism, including age, diabetes, BMI, and surgical laterality. Notably, elderly patients (>65 years) represented a higher proportion of those who developed hypothyroidism at 1 month (31.11%), followed closely by patients aged 56–65 years (29.54%). These findings align with the general understanding that older adults are more susceptible to thyroid dysfunction. However, in our adjusted Cox model, age was not a statistically significant predictor of the timing of hypothyroidism, suggesting that other factors may play a stronger role in determining outcomes. This elevated risk may be attributable to age-related declines in thyroid reserve, which limit the compensatory capacity of the remaining thyroid lobe [15]. The fact that the incidence of hypothyroidism persisted in older adults at 6 and 12 months further emphasizes the need for vigilant thyroid function monitoring in this vulnerable group. Although unadjusted analyses showed higher hypothyroidism rates in patients >65 years, this age group was not a statistically significant predictor of the timing of hypothyroidism in our adjusted Cox model. This may reflect the influence of other variables, such as BMI or surgical laterality, on thyroid function outcomes. Age >65 years might still have represented a significant risk factor if hypothyroidism were analyzed as a binary outcome, highlighting the need for future research to explore this relationship using alternative analytical approaches.

In addition to age, our analysis revealed a significant association between elevated BMI and hypothyroidism risk. Patients classified as severely obese exhibited higher rates of hypothyroidism at all time points compared to their normal-weight counterparts. That being said, in our adjusted Cox model, overweight (compared to normal weight individuals) was deemed a significant factor for higher risk of hypothyroidism, while severe obesity was deemed insignificant. This finding is consistent with the existing literature, suggesting a complex interplay between obesity, metabolic function, and thyroid health [16,17]. The increased risk in these populations may be related to altered thyroid hormone metabolism, resistance, or reduced sensitivity to feedback mechanisms that regulate TSH levels [18,19].

Obesity is known to correlate with elevated TSH levels, but this does not always indicate true hypothyroidism [20]. Our findings reinforce this distinction, as BMI was not an independent predictor of the timing of hypothyroidism in our adjusted model. This highlights the importance of carefully interpreting thyroid function in obese patients to avoid overdiagnosis and unnecessary treatment.

Laterality of surgery did not significantly affect hypothyroidism rates at 1 or 6 months. However, at 12 months, patients undergoing left-sided hemithyroidectomy demonstrated a significantly higher incidence of hypothyroidism compared to those with right-sided hemithyroidectomy. This finding suggests that surgical laterality may influence long-term thyroid outcomes, warranting further investigation into the potential anatomical or physiological factors involved. That being said, after adjusting for potential confounders in the adjusted Cox model, laterality was deemed an insignificant determinant of time to hypothyroidism.

### 4.1. Clinical Implications

Our findings have several important clinical implications. First, given the substantial incidence of early postoperative hypothyroidism, it is crucial to implement structured follow-up protocols that include routine thyroid function testing, particularly during the first year after surgery. Second, patients at higher risk, such as the elderly and those with elevated BMI, may benefit from more frequent monitoring and early consideration of thyroid hormone supplementation, even in the absence of overt clinical symptoms. Identifying and treating subclinical hypothyroidism in these populations could help to mitigate potential complications, including cardiovascular morbidity and impaired quality of life. Moreover, the association between surgical laterality and hypothyroidism highlights the need for a deeper understanding of the anatomical and physiological factors that may influence thyroid function after hemithyroidectomy. Although the lack of separate analyses for subclinical and overt hypothyroidism is a limitation, our findings still emphasize the need for routine thyroid function monitoring post-hemithyroidectomy. Future research should focus on whether tailored surgical approaches or postoperative interventions can reduce the risk of hypothyroidism in high-risk patients.

### 4.2. Study Strengths and Limitations

This study has several strengths, including its large sample size and comprehensive follow-up period, which enabled us to capture both early and late thyroid dysfunction. Additionally, our detailed stratification by patient characteristics allowed for the identification of key risk factors that can inform clinical practice. However, several limitations should be acknowledged. First, the retrospective nature of the study may introduce selection bias, and the reliance on electronic medical records for symptom reporting could result in underestimation of overt hypothyroidism. Second, this study considered both subclinical and overt hypothyroidism under the broader term ‘hypothyroidism’, without performing separate analyses for these categories. While this approach simplifies interpretation, it limits our ability to provide specific insights into the unique risk factors and outcomes associated with each subtype. Third, our dataset did not include information on thyroid antibody levels (e.g., TPO and thyroglobulin antibodies) or autoimmune thyroid disease diagnoses. This limitation reduces our ability to assess whether postoperative hypothyroidism was directly related to hemithyroidectomy or pre-existing autoimmune thyroid dysfunction. Including such data in future studies will help to delineate these relationships and strengthen the findings. Additionally, the generalizability of our findings may be limited to similar tertiary care settings, and future prospective studies are needed to validate our results in more diverse populations.

## 5. Conclusions

Hypothyroidism is a common and clinically significant complication following hemithyroidectomy, particularly in elderly and overweight patients. The high incidence of thyroid dysfunction within the first year postoperatively underscores the importance of routine thyroid function monitoring and individualized management strategies. Future studies should aim to further elucidate the pathophysiology of post-hemithyroidectomy hypothyroidism and explore interventions to reduce its incidence, ultimately improving patient outcomes.

## Figures and Tables

**Figure 1 jcm-14-00919-f001:**
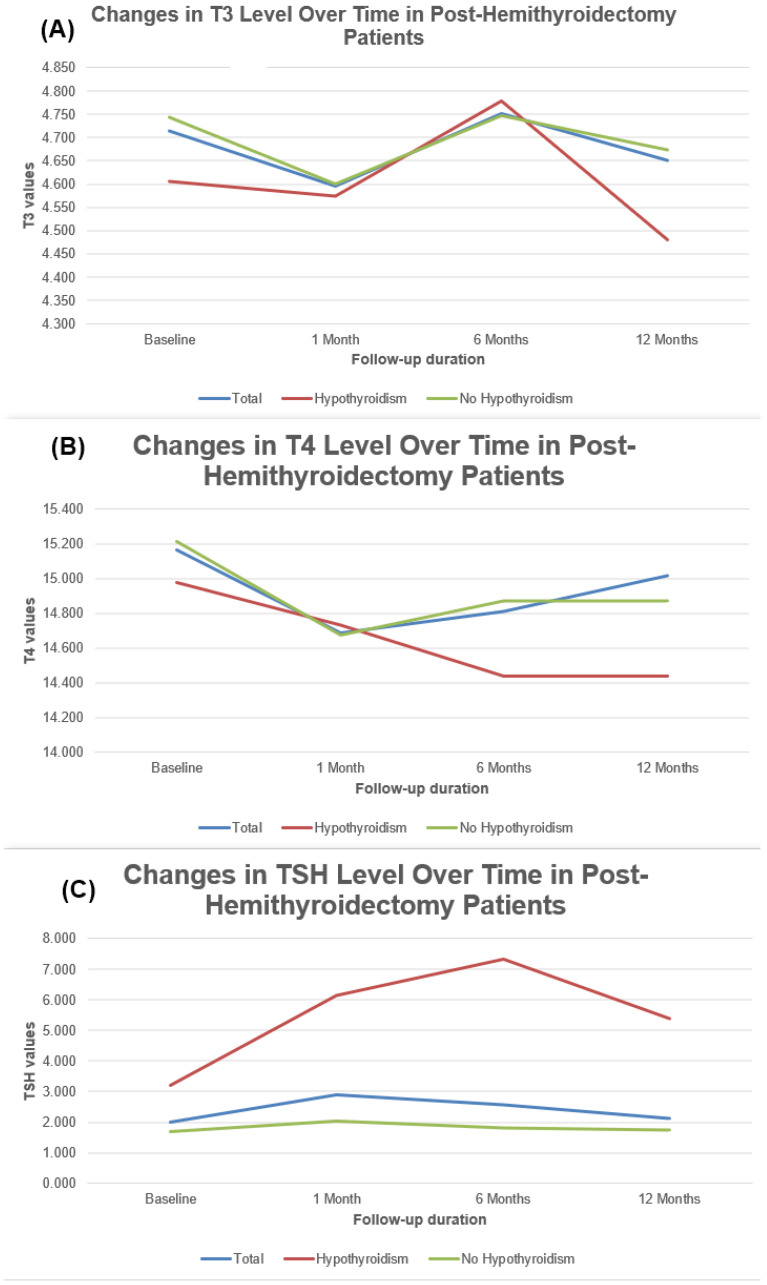
Temporal changes in (**A**) T3, (**B**) T4, and (**C**) TSH levels over time, from baseline to 12 months postoperatively.

**Figure 2 jcm-14-00919-f002:**
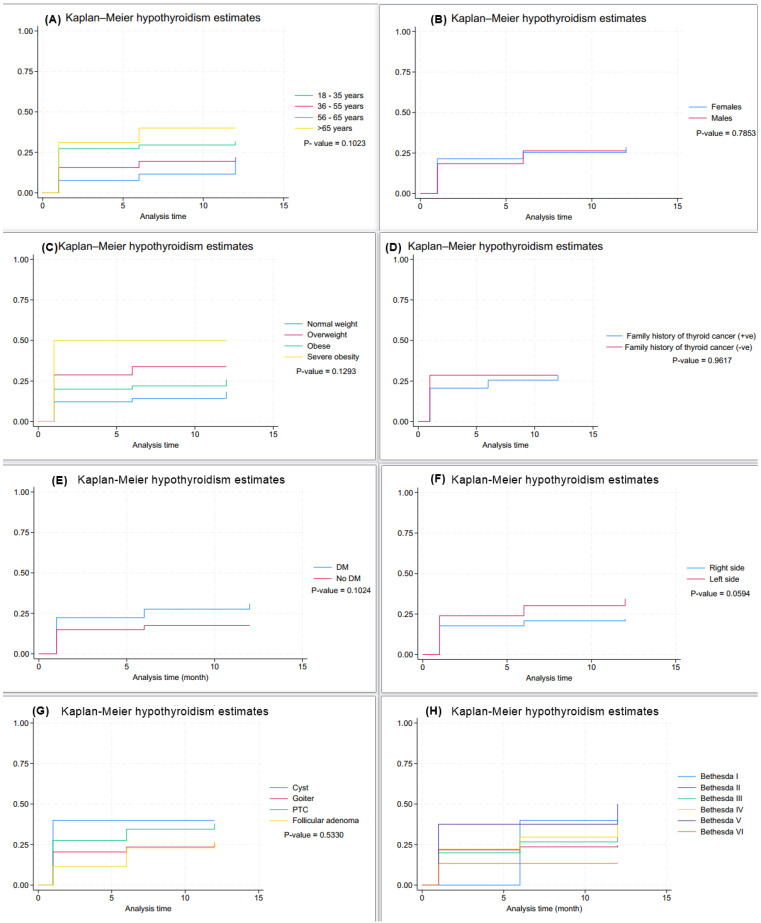
Kaplan–Meier curves of time to post-hemithyroidectomy hypothyroidism, according to (**A**) patients’ age, (**B**) gender, (**C**) BMI, (**D**) family history of thyroid cancer, (**E**) DM, (**F**) laterality, (**G**) hemithyroidectomy indication, and (**H**) Bethesda class.

**Table 1 jcm-14-00919-t001:** Baseline clinicodemographic characteristics of included patients post-hemithyroidectomy.

Variable	Mean (SD) or Frq. (%)
Age—yr	
Mean (SD)	53.15 (14.33)
Distribution—no. (%)	
Young adult (18–35 yr)	26 (13.54%)
Middle-aged adult (36–55 yr)	77 (40.10%)
Older adults (56–65 yr)	44 (22.92%)
Elderly (>65 yr)	45 (23.44%)
Gender—no. (%)	
Male	38 (19.79%)
Female	154 (80.21%)
Body mass index	
Mean (SD)	28.32 (5.96)
Distribution—no. (%)	
Normal weight (18.5–24.9)	49 (29.52%)
Overweight (25–29.9)	59 (35.54%)
Obese (30–39.9)	50 (30.12%)
Severe obesity (>40)	8 (4.82%)
Missing values	22 (13.54%)
Family history of thyroid cancer—no. (%)	
No	184 (96.34%)
Yes	7 (3.66%)
Laterality/Operative side—no. (%)	
Right	96 (50%)
Left	96 (50%)
Indication for surgery—no. (%)	
Cyst	5 (2.6%)
Goiter	132 (68.75%)
PTC	29 (15.10%)
Follicular adenoma	26 (13.54%)
Diabetes mellitus—no. (%)	
No	152 (79.17%)
Yes	40 (20.83%)
Thyroid cytopathology: Bethesda—no. (%)	
I	5 (2.62%)
II	106 (55.50%)
III	30 (15.71%)
IV	27 (14.14%)
V	8 (4.19%)
VI	15 (7.85%)
Preoperative thyroid function	
T3 level	
Mean (SD)	4.71 (0.64)
T4 level	
Mean (SD)	15.17 (1.94)
TSH level	
Mean (SD)	2.00 (1.26)

yr: year; TSH: thyroid-stimulating hormone; SD: standard deviation; no.: number of patients; Frq: frequency.

**Table 2 jcm-14-00919-t002:** The mean changes (95% confidence interval) in T3, T4, and TSH levels over time post-hemithyroidectomy.

		Preoperative	1 Month	6 Months	12 Months
**T3 Level**	1 Month	−0.12 (−0.26, 0.02)	-	-	-
	*p*-value	0.092	-	-	-
	6 Months	0.04 (0.17, −0.09)	0.16 (0.29, 0.01)	-	-
	*p*-value	0.582	0.034 ^a^	-	-
	12 Months	−0.06 (0.08, −0.21)	0.06 (0.21, −0.09)	−0.09 (0.05, −0.25)	-
	*p*-value	0.409	0.472	0.200	-
**T4 Level**		**Preoperative**	**1 Month**	**6 Months**	**12 Months**
	1 Month	−0.48 (−0.89, −0.07)	-	-	-
	*p*-value	0.023 ^a^	-	-	-
	6 Months	−0.35 (0.02, −0.73)	−0.13 (−0.53, 0.27)	-	-
	*p*-value	0.064	0.532	-	-
	12 Months	−0.15 (0.22, −0.52)	0.33 (0.73, −0.06)	0.21 (0.56, −0.15)	-
	*p*-value	0.436	0.100	0.257	-
**TSH Level**		**Preoperative**	**1 Month**	**6 Months**	**12 Months**
	1 Month	0.90 (0.54, 1.25)	-	-	-
	*p*-value	0.000 ^a^	-	-	-
	6 Months	0.58 (1.12, 0.04)	−0.32 (0.28, −0.92)	-	-
	*p*-value	0.036 ^a^	0.293	-	-
	12 Months	0.13 (0.41, −0.14)	−0.77 (−0.39, −1.14)	−0.44 (0.11, −1.00)	-
	*p*-value	0.341	0.000 ^a^	0.112	-

^a^ statistically significant difference (*p* < 0.05). The data are presented as the mean difference between timepoints (95%confidence interval). TSH: thyroid-stimulating hormone.

**Table 3 jcm-14-00919-t003:** The incidence rate of post-hemithyroidectomy hypothyroidism, stratified by follow-up time and patients’ characteristics.

	1 Month			6 Months			12 Months		
	No (N = 151)	Yes (N = 41)	*p*	No (N = 165)	Yes (N = 27)	*p*	No (N = 171)	Yes (N = 21)	*p*
Age—no. (%)									
18–35 yr	24 (92.31%)	2 (7.69%)	0.033	24 (92.31%)	2 (7.69%)	0.013	22 (84.62%)	4 (15.38%)	0.237
36–55 yr	65 (84.42%)	12 (15.58%)		69 (89.61%)	8 (10.39%)		71 (92.21%)	6 (7.79%)	
56–65 yr	31 (70.46%)	13 (29.54%)		40 (90.91%)	4 (9.09%)		41 (93.19%)	3 (6.81%)	
>65 yr	31 (68.89%)	14 (31.11%)		32 (71.11%)	13 (28.89%)		37 (82.22%)	8 (17.78%)	
Gender—no. (%)									
Female	120 (77.93%)	34 (22.07%)	0.622	133 (86.37%)	21 (13.63%)	0.732	135 (87.66%)	19 (12.34%)	0.211
Male	31 (81.58%)	7 (18.42%)		32 (84.21%)	6 (15.79%)		36 (94.74%)	2 (5.26%)	
BMI									
Normal weight	42 (85.72%)	7 (14.28%)	0.079	44 (89.80%)	5 (10.20%)	0.429	41 (83.67%)	8 (16.33%)	0.511
Overweight	42 (71.19%)	17 (28.81%)		50 (84.75%)	9 (15.25%)		54 (91.53%)	5 (8.47%)	
Obese	40 (80%)	10 (20%)		46 (92%)	4 (8%)		46 (92%)	4 (8%)	
Severe Obesity	4 (50%)	4 (50%)		6 (75%)	2 (25%)		7 (87.50%)	1 (12.50%)	
Family history of thyroid cancer—no. (%)									
No	145 (78.81%)	39 (21.19%)	0.641	158 (85.87%)	26 (14.13%)	0.991	163 (88.59%)	21 (11.41%)	0.343
Yes	5 (71.437%)	2 (28.57%)		6 (85.71%)	1 (14.29%)		7 (100%)	0 (0%)	
Laterality—no. (%)									
Right	79 (82.30%)	17 (17.70%)	0.218	84 (87.50%)	12 (12.50%)	0.533	90 (93.75%)	6 (6.25%)	0.037
Left	72 (75%)	24 (25%)		81 (84.38%)	15 (15.62%)		81 (84.37%)	15 (15.63%)	
Surgical indication—no. (%)									
Cyst	3 (60%)	2 (40%)	0.362	3 (60%)	2 (40%)	0.087	4 (80%)	1 (20%)	0.928
Goiter	104 (78.79%)	28 (21.21%)		118 (89.39%)	14 (10.61%)		118 (89.39%)	14 (10.61%)	
PTC	21 (72.42%)	8 (27.58%)		22 (75.86%)	7 (25.9324.14%)		26 (89.66%)	3 (10.34%)	
Follicular adenoma	23 (88.47%)	3 (11.53%)		22 (84.62%)	4 (15.38%)		23 (88.46%)	3 (11.54%)	
Diabetes mellitus—no. (%)									
No	117 (76.97%)	35 (23.03%)	0.27	130 (85.53%)	22 (14.47%)	0.749	133 (87.50%)	19 (12.50%)	0.176
Yes	34 (85%)	6 (15%)		35 (87.50%)	5 (12.50%)		38 (95%)	2 (5%)	
Bethesda—no. (%)									
I	5 (100%)	0 (0%)	0.652	3 (60%)	2 (40%)	0.435	4 (80%)	1 (20%)	0.782
II	82 (77.36%)	24 (22.64%)		94 (88.68%)	12 (11.32%)		94 (88.77%)	12 (11.23%)	
III	24 (80%)	6 (20%)		25 (83.33%)	5 (16.67%)		26 (86.67%)	4 (13.33%)	
IV	21 (78.57%)	6 (21.43%)		22 (81.48%)	5 (18.52%)		24 (88.89%)	3 (11.11%)	
V	5 (62.50%)	3 (37.50%)		7 (87.50%)	1 (12.50%)		7 (87.50%)	1 (12.50%)	
VI	13 (86.67%)	2 (13.33%)		14 (93.33%)	1 (6.67%)		15 (100%)	0 (0%)	

no.: number of patients; N: number of patients in each group; BMI: body mass index.

**Table 4 jcm-14-00919-t004:** Unadjusted and adjusted Cox proportional hazards models for determinants of post-hemithyroidectomy hypothyroidism.

	Unadjusted Cox Proportional Hazards Model						Adjusted Cox Proportional Hazards Model					
	HR	SE	Z	*P*	Low CI	High CI	aHR	SE	Z	*P*	Low CI	High CI
Age												
18–35 yr	Reference	-	-	-	-	-	-	-	-	-	-	-
36–55 yr	1.179	0.600	0.320	0.747	0.435	3.195	0.730	0.404	−0.570	0.570	0.247	2.158
56–65 yr	1.765	0.920	1.090	0.276	0.635	4.901	1.244	0.697	0.390	0.696	0.415	3.727
>65 yr	2.269	1.148	1.620	0.105	0.842	6.115	1.500	0.857	0.710	0.478	0.490	4.594
Gender												
Female	Reference	-	-	-	-	-	-	-	-	-	-	-
Male	0.916	0.321	−0.250	0.803	0.461	1.821	----------------Excluded----------------					
Body mass index												
Normal weight	Reference	-	-	-	-	-	-	-	-	-	-	-
Overweight	1.935	0.777	1.640	0.100	0.881	4.252	2.368	1.023	2.000	0.046 ^a^	1.016	5.523
Obese	1.444	0.626	0.850	0.397	0.617	3.379	1.459	0.676	0.810	0.415	0.588	3.617
Severe Obesity	3.007	1.809	1.830	0.067	0.925	9.780	3.185	2.044	1.810	0.071	0.906	11.201
Family history of thyroid cancer												
No family history	Reference	-	-	-	-	-	-	-	-	-	-	-
Positive family history	1.032	0.744	0.040	0.965	0.251	4.237	----------------Excluded----------------					
Laterality												
Right side	Reference	-	-	-	-	-	-	-	-	-	-	-
Left side	1.611	0.450	1.710	0.088	0.932	2.785	1.682	0.534	1.640	0.102	0.903	3.134
Surgical indication												
Goiter	Reference	-	-	-	-	-	-	-	-	-	-	-
PTC	1.516	0.526	1.200	0.230	0.768	2.993	1.114	0.444	0.270	0.787	0.510	2.431
Follicular adenoma	1.025	0.426	0.060	0.952	0.455	2.313	1.026	0.473	0.060	0.955	0.416	2.534
Cyst	1.655	1.205	0.690	0.489	0.398	6.893	0.000	0.000	0.000	1.000	0.000	.
Diabetes mellitus												
Negative	Reference	-	-	-	-	-	-	-	-	-	-	-
Positive	0.550	0.223	−1.470	0.140	0.249	1.218	0.379	0.183	−2.010	0.044 ^a^	0.147	0.975
Bethesda												
I	Reference	-	-	-	-	-	-	-	-	-	-	-
II	0.626	0.459	−0.640	0.523	0.148	2.638	----------------Excluded----------------					
III	0.766	0.599	−0.340	0.733	0.165	3.546	----------------Excluded----------------					
IV	0.953	0.739	−0.060	0.951	0.209	4.352	----------------Excluded----------------					
V	1.340	1.161	0.340	0.735	0.245	7.322	----------------Excluded----------------					
VI	0.331	0.331	−1.110	0.268	0.047	2.347	----------------Excluded----------------					

^a^ statistically significant risk factor (*p* < 0.05). HR: hazard ratio; aHR: adjusted hazard ratio; SE: standard error; CI: confidence interval; *P*: *p*-value; yr: year;- empty data

## Data Availability

The dataset analyzed in this manuscript can be provided by the corresponding author upon reasonable request.

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
