# Peer review of "Hypothyroidism After Hemithyroidectomy: A Retrospective Analysis of Temporal Trends and Key Risk Factors"

_jcm, 2025, doi:10.3390/jcm14030919_

Round 1

Reviewer 1 Report

Comments and Suggestions for Authors

 I find many concerns in the manuscript in its current form.  There would need to be substantial revisions to make this scientifically sound.   The main issue with the manuscript is the lack of clear definition of sub-clinical vs overt hypothyroidism.  Per the ATA/AACE guidelines in 2012 subclinical hypothyroidism is an elevated TSH with normal T4 whereas a TSH >10 combined with a low free T4 constitutes overt hypothyroidism.  The use of elevated TSH with clinical symptoms of fatigue, weight gain, and depression does not fit into these definitions.  Additionally the authors do not state if they are considering both subacute and overt hypothyroidism in the population of "hypothyroidism."  They did not state what thyroid function tests prompted thyroid hormone replacement therapy.

Although it is stated that thyroid function was normal prior to thyroidectomy, the antibody status (TPO and thyroglobulin Ab) was not stated.  Autoimmune thyroid disease is quite common and predisposes to hypothyroidism, so this would be important to know both preoperatively and postoperatively as the cause of hypothyroidism could presumably be unrelated to the surgery.  Autoimmune thyroid disease is particularly prevalent in females as well and the population of this study included more females.

Hypothyroidism is more common in patients >60 years old.  This point must be made and comparison to the general population is necessary here.  Also, there is clear literature about increase in TSH levels in obesity, which is NOT hypothyroidism - I do not think this is clear in the paper and should be more clearly addressed as a pitfall in the conclusion.

There is a spelling error in figure 1 A, B, C (Timme instead of Time).

Author Response

REVIEWER #1 
Comment 1: I find many concerns in the manuscript in its current form.  There would need to be substantial revisions to make this scientifically sound.   The main issue with the manuscript is the lack of clear definition of sub-clinical vs overt hypothyroidism.  Per the ATA/AACE guidelines in 2012 subclinical hypothyroidism is an elevated TSH with normal T4 whereas a TSH >10 combined with a low free T4 constitutes overt hypothyroidism.  The use of elevated TSH with clinical symptoms of fatigue, weight gain, and depression does not fit into these definitions.  Additionally the authors do not state if they are considering both subacute and overt hypothyroidism in the population of "hypothyroidism."  They did not state what thyroid function tests prompted thyroid hormone replacement therapy.
Response: Thank you for your insightful feedback. Your comments were immensely helpful in driving meaningful conclusions and providing us with more gaps to address in our future work.
We used the ATA/AACE guidelines cutoff points to determine subclinical and overt hypothyroidism; however, as per insitutional protocols, thyroid hormone replacement therapy was initiated depending on the prior criteria, while putting into account clinical presentation (i.e., fatigue, weight gain, depression).
We updated this part accordingly in the manuscript in the methods section to avoid any confusion to the reader as follows: "Subclinical hypothyroidism was defined as a serum thyroid-stimulating hormone (TSH) level 4.20-10 µIU/mL with normal free T4 levels in asymptomatic patients, as per ATA/AACE 2012 guidelines [reference]. Overt hypothyroidism was defined by elevated TSH levels  (>10 µIU/mL) combined with a low free T4 level. Both conditions were considered as “hypothyroidism” for this study. Thyroid hormone replacement therapy was initiated in patients with TSH >10 µIU/mL or those with a low free T4, particularly if accompanied by clinical symptoms such as fatigue, weight gain, or depression. These thresholds were based on clinical guidelines and institutional protocols. "
Additionally, under the "Clinical implications" part, we added this sentence: "Although the lack of separate analyses for subclinical and overt hypothyroidism is a limitation, our findings still emphasize the need for routine thyroid function monitoring post-hemithyroidectomy."
We also acknolwedged the drawback of not analyzing subclinical and overt hypothyroidism separately in the Limitations section as follows: "This study considered both subclinical and overt hypothyroidism under the broader term 'hypothyroidism' without performing separate analyses for these categories. While this approach simplifies interpretation, it limits our ability to provide specific insights into the unique risk factors and outcomes associated with each subtype."

Comment 2: Although it is stated that thyroid function was normal prior to thyroidectomy, the antibody status (TPO and thyroglobulin Ab) was not stated.  Autoimmune thyroid disease is quite common and predisposes to hypothyroidism, so this would be important to know both preoperatively and postoperatively as the cause of hypothyroidism could presumably be unrelated to the surgery.  Autoimmune thyroid disease is particularly prevalent in females as well and the population of this study included more females.
Response: Thank you for raising this point. Unfortunately, our dataset did not include information on TPO or thyroglobulin antibody levels or diagnoses of autoimmune thyroid disease. We acknowledge that this is a limitation of our study, as it limits our ability to fully assess the contribution of autoimmune thyroid disease to postoperative hypothyroidism. We have added this limitation to the manuscript as follows: "Our dataset did not include information on thyroid antibody levels (e.g., TPO and thyroglobulin antibodies) or autoimmune thyroid disease diagnoses. This limitation reduces our ability to assess whether postoperative hypothyroidism was directly related to hemithyroidectomy or preexisting autoimmune thyroid dysfunction. Including such data in future studies will help delineate these relationships and strengthen the findings."

Comment 3: Hypothyroidism is more common in patients >60 years old.  This point must be made and comparison to the general population is necessary here.  Also, there is clear literature about increase in TSH levels in obesity, which is NOT hypothyroidism - I do not think this is clear in the paper and should be more clearly addressed as a pitfall in the conclusion.
Response: Thank you for your thoughtful feedback regarding the roles of age and BMI in post-hemithyroidectomy hypothyroidism. 
Our unadjusted analyses demonstrated that patients >65 years old exhibited higher rates of hypothyroidism, with 34.15% developing hypothyroidism at 1 month postoperatively. However, in our adjusted Cox proportional hazards model, which accounted for potential confounders such as BMI and surgical laterality, age >65 years was not a statistically significant predictor of the "time to hypothyroidism" (adjusted HR 1.500, 95% CI [0.490, 4.594], p = 0.478). This finding suggests that other variables may have played a more prominent role in determining the timing of thyroid dysfunction. We acknowledge that age may have shown statistical significance if hypothyroidism were analyzed as a binary outcome rather than a time-to-event variable. This limitation has been noted in the manuscript, and we have emphasized the importance of exploring this relationship in future studies using alternative analytical approaches as follows: "Although unadjusted analyses showed higher hypothyroidism rates in patients >65 years, this age group was not a statistically significant predictor of the timing of hypothyroidism in our adjusted Cox model. This may reflect the influence of other variables, such as BMI or surgical laterality, on thyroid function outcomes. Age >65 years might still represent a significant risk factor if hypothyroidism were analyzed as a binary outcome, highlighting the need for future research to explore this relationship using alternative analytical approaches."
With respect to BMI, obesity is known to correlate with elevated TSH levels, which are not always indicative of true hypothyroidism. Our Cox proportional hazards model adjusted for BMI categories and found that while severe obesity showed a trend toward increased risk (adjusted HR 3.185, 95% CI [0.906, 11.201]), it was not statistically significant (p = 0.071). This finding highlights the distinction between obesity-related TSH elevations and thyroid dysfunction, which is important for interpreting thyroid function tests in clinical practice. We have revised the manuscript to clarify this distinction and included it as a limitation and a direction for future research as follows: "Obesity is known to correlate with elevated TSH levels, but this does not always indicate true hypothyroidism. Our findings reinforce this distinction, as BMI was not an independent predictor of the timing of hypothyroidism in our adjusted model. This highlights the importance of carefully interpreting thyroid function in obese patients to avoid overdiagnosis and unnecessary treatment."

Comment 4: There is a spelling error in figure 1 A, B, C (Timme instead of Time).
Response: Thank you for noticing this typo mistake. We corrected it and attached the corrected Figure.

Reviewer 2 Report

Comments and Suggestions for Authors

1.     Data Errors

Several significant data errors need careful verification and correction:

 (1) In line 23, you mention the rate of hypothyroidism in elderly patients (>65 years) as 34.15% at 1 month. However, according to Table 3, among all elderly patients, 14 developed hypothyroidism 1 month after surgery, while 31 did not. Therefore, the rate of hypothyroidism in elderly patients is 14/(14+31) = 31.11%, not 34.15%. The 34.15% instead represents the proportion of elderly patients among all 41 patients who developed hypothyroidism at 1 month after surgery (14/41 = 34.15%). Please clarify this discrepancy.

 (2) In lines 181-183, you state that at 1 month, 41 patients (21.35%) developed hypothyroidism, with the elderly (>65 years) showing a significantly higher incidence compared to younger age groups (34.15% vs. 15.89%, p = 0.033). However, the 34.15% figure refers to the percentage of elderly patients among all those who developed hypothyroidism in the first month. Additionally, the 15.89% figure represents the proportion of young patients (18-35 years) among all 151 patients who did not develop hypothyroidism (24/151 = 15.89%). These two percentages reflect different meanings and should not be compared directly.

 (3) In lines 190-193, you state that by 12 months, patients who underwent left-sided hemithyroidectomy had a higher incidence of hypothyroidism compared to those with right-sided surgery (28.57% vs. 52.63%, p = 0.037). However, according to Table 3, the incidence of hypothyroidism in the left-sided group is 15/(15+81) = 15.63%, and in the right-sided group is 6/(90+6) = 6.25%.

 (4) In lines 23-25, the hazard ratio of 2.368 pertains to patients who are overweight (BMI 25-29.9). However, you incorrectly associate this figure with patients classified as severely obese (BMI >40). Furthermore, you omit the hazard ratio for the severely obese group, which is not statistically significant according to the results in Table 4. Please correct this and clarify the findings.

 2.     Patients with Thyroiditis

Patients with thyroiditis are likely to develop hypothyroidism. Did you check patients' anti-thyroglobulin antibody (ATA) and anti-thyroid peroxidase antibody (Anti-TPO) levels? Did you include patients with high ATA or Anti-TPO levels but normal TSH, T3, and T4 values in your analysis? Clarification on this inclusion criterion is essential.

Author Response

REVIEWER #2
1.     Data Errors
Comment (1): In line 23, you mention the rate of hypothyroidism in elderly patients (>65 years) as 34.15% at 1 month. However, according to Table 3, among all elderly patients, 14 developed hypothyroidism 1 month after surgery, while 31 did not. Therefore, the rate of hypothyroidism in elderly patients is 14/(14+31) = 31.11%, not 34.15%. The 34.15% instead represents the proportion of elderly patients among all 41 patients who developed hypothyroidism at 1 month after surgery (14/41 = 34.15%). Please clarify this discrepancy.
Response: Thank you for highlighting this very important point. We have carefully updated the corresponding section of the manuscript and Table 3 to address the concerns raised. The data has been revised and recalculated as per your suggestions to ensure consistency and precision. 

We have done the following revisions:
Results section "3.3. Temporal trends of post-hemithyroidectomy hypothyroidism" as follows: 
we added: "At 1 month postoperatively, 41 patients (21.35%) developed hypothyroidism, while 151 patients (78.65%) remained euthyroid. Among those who developed hypothyroidism, the incidence was highest in patients >65 years (31.11%), followed by those aged 56–65 years (29.54%). In contrast, younger patients aged 18–35 years had the lowest incidence (7.69%). At 6 months, 27 patients (14.06%) were hypothyroid, with a continued higher incidence in older age groups. By 12 months, 21 patients (10.94%) remained hypothyroid, with the elderly (>65 years) showing a rate of 17.78%. This temporal trend highlights that older age groups consistently exhibit higher hypothyroidism rates compared to younger cohorts.
Regarding BMI, patients with severe obesity demonstrated the highest incidence of hypothyroidism across all follow-up periods, with rates of 50.00% at 1 month, 25.00% at 6 months, and 12.50% at 12 months. Patients with normal weight exhibited a lower incidence (14.28% at 1 month). However, the association between BMI and hypothyroidism was not statistically significant (p > 0.05 at all time points).
No significant differences were observed in hypothyroidism incidence based on gender or family history of thyroid cancer. Surgical laterality, however, showed a significant difference at 12 months, with patients undergoing left-sided hemithyroidectomy having a higher incidence of hypothyroidism (15.63%) compared to those with right-sided hemithyroidectomy (6.25%; p = 0.037)."

Abstract:
we removed: "At 1 month postoperatively, with ........"
we added: "At 1 month postoperatively, with the highest incidence observed in elderly patients (>65 years; 31.11%) and those aged 56–65 years (29.54%), compared to younger age groups (7.69% in patients aged 18–35 years). The incidence decreased over time, with 14.06% and 10.94% of patients being hypothyroid at 6 and 12 months, respectively. Severe obesity was associated with the highest hypothyroidism rates across all follow-up periods, particularly at 1 month (50.00%)." plus "Patients undergoing left-sided hemithyroidectomy had a higher incidence of hypothy-roidism at 12 months compared to right-sided patients (15.63% vs. 6.25%, p=0.037). "

Discussion:
(1) we removed: "Notably, elderly patients (>65 years) were at significantly higher risk of developing hypothyroidism, particularly in the early postoperative period, with a 34.15% incidence at 1 month."
Instead, we added: "Notably, elderly patients (>65 years) represented a higher proportion of those who developed hypothyroidism at 1 month (31.11%), followed closely by patients aged 56–65 years (29.54%). These findings align with the general understanding that older adults are more susceptible to thyroid dysfunction. However, in our adjusted Cox model, age was not a statistically significant predictor of the timing of hypothyroidism, suggesting that other factors may play a stronger role in determining outcomes."

(2) we removed: "Surgical laterality also emerged as a notable risk factor, with patients undergoing left-sided hemithyroidectomy showing a higher incidence of hypothyroidism at 12 months compared to those who had right-sided surgery. The reasons for this discrepancy remain unclear, although anatomical variations in the vascular and neural supply to the thyroid lobes could play a role. Further studies are warranted to elucidate the mechanisms underlying this association."
Instead, we added: "Laterality of surgery did not significantly affect hypothyroidism rates at 1 or 6 months. However, at 12 months, patients undergoing left-sided hemithyroidectomy demonstrated a significantly higher incidence of hypothyroidism compared to those with right-sided hemithyroidectomy (15.63% vs. 6.25%; p = 0.037). This finding suggests that surgical laterality may influence long-term thyroid outcomes, warranting further investigation into potential anatomical or physiological factors involved. That being said, after adjusting for potential confounders in the adjusted Cox model, laterality was deemed an insignificant determinant of time to hypothyroidism."

(3) we added: "Patients classified as severely obese exhibited higher rates of hypothyroidism at all time points compared to their normal-weight counterparts. That being said, in our adjusted Cox model, overweight (compared to normal weight individuals) was deemed significant of higher risk of hypothyroidism, while severe obesity was deemed insignificant."

Comment (2) In lines 181-183, you state that at 1 month, 41 patients (21.35%) developed hypothyroidism, with the elderly (>65 years) showing a significantly higher incidence compared to younger age groups (34.15% vs. 15.89%, p = 0.033). However, the 34.15% figure refers to the percentage of elderly patients among all those who developed hypothyroidism in the first month. Additionally, the 15.89% figure represents the proportion of young patients (18-35 years) among all 151 patients who did not develop hypothyroidism (24/151 = 15.89%). These two percentages reflect different meanings and should not be compared directly.
Response: Thank you. Please check our response to the above comment.

Comment (3) In lines 190-193, you state that by 12 months, patients who underwent left-sided hemithyroidectomy had a higher incidence of hypothyroidism compared to those with right-sided surgery (28.57% vs. 52.63%, p = 0.037). However, according to Table 3, the incidence of hypothyroidism in the left-sided group is 15/(15+81) = 15.63%, and in the right-sided group is 6/(90+6) = 6.25%.
Response: Thank you. Please check our response to the above comment.

Comment (4) In lines 23-25, the hazard ratio of 2.368 pertains to patients who are overweight (BMI 25-29.9). However, you incorrectly associate this figure with patients classified as severely obese (BMI >40). Furthermore, you omit the hazard ratio for the severely obese group, which is not statistically significant according to the results in Table 4. Please correct this and clarify the findings.
Response: Thank you for your great attention to detail! We appreciate your help in revising our manuscript.
We removed all respective parts of the manuscript where severe obesity was linked with significant risk of hypothyroidism (from Abstract to Discussion).
Instead, we updated these data in the following sections:
(1) Figure 2 Legend as follows: "In the adjusted model, being overweight (HR 2.368, 95% CI [1.016, 5.523], p = 0.046) significantly increased the risk of hypothyroidism, while severe obesity was not a sig-nificant predictor (HR 3.185, 95% CI [0.906, 11.201], p = 0.071)."
(2) Discussion section: "Patients classified as severely obese exhibited higher rates of hypothyroidism at all time points compared to their normal-weight counterparts. That being said, in our adjusted Cox model, overweight (compared to normal weight individuals) was deemed significant of higher risk of hypothyroidism, while severe obesity was deemed insignificant."

 2.     Patients with Thyroiditis
Comment: Patients with thyroiditis are likely to develop hypothyroidism. Did you check patients' anti-thyroglobulin antibody (ATA) and anti-thyroid peroxidase antibody (Anti-TPO) levels? Did you include patients with high ATA or Anti-TPO levels but normal TSH, T3, and T4 values in your analysis? Clarification on this inclusion criterion is essential.
Response: Thank you for your question. This was a similar question by the other reviewer as well. Thank you for raising this point. Unfortunately, our dataset did not include information on TPO or thyroglobulin antibody levels or diagnoses of autoimmune thyroid disease. We acknowledge that this is a limitation of our study, as it limits our ability to fully assess the contribution of autoimmune thyroid disease to postoperative hypothyroidism. We have added this limitation to the manuscript as follows: "Our dataset did not include information on thyroid antibody levels (e.g., TPO and thyroglobulin antibodies) or autoimmune thyroid disease diagnoses. This limitation reduces our ability to assess whether postoperative hypothyroidism was directly related to hemithyroidectomy or preexisting autoimmune thyroid dysfunction. Including such data in future studies will help delineate these relationships and strengthen the findings."

Round 2

Reviewer 2 Report

Comments and Suggestions for Authors

The authors have addressed the queries satisfactorily, and I have no additional questions.